# Adversarial Robustness Evaluation of Deep Learning Segmentation Models and Loss Functions in Prostate MRI

**Kosmas K. Apostolidis**
*Biomedical Research Institute*
*FORTH*
Ioannina, Greece
k.apostolidis@uoi.gr

**Dimitrios I. Zaridis**
*Biomedical Engineering Laboratory*
*School of Electrical & Computer Engineering*
*National Technical University of Athens and FORTH*
Athens & Ioannina, Greece
dimizaridis@mail.ntua.gr

**Nikolaos S. Tachos**
*Biomedical Research Institute*
*FORTH*
Ioannina, Greece
ntachos@bri.forth.gr

**Vasileios C. Pezoulas**
*Biomedical Research Institute*
*FORTH*
Ioannina, Greece
vpezoulas@bri.forth.gr

**Kostas Marias**
*Computational BioMedicine Laboratory*
*Institute of Computer Science - FORTH*
Heraklion, Greece
kmarias@ics.forth.gr

**Manolis Tsiknakis**
*Computational BioMedicine Laboratory*
*Institute of Computer Science - FORTH*
Heraklion, Greece
tsiknaki@ics.forth.gr

**Dimitrios I. Fotiadis**
*Unit of Medical Technology and Intelligent Information Systems and*
*Biomedical Research Institute*
*University of Ioannina and FORTH*
Ioannina, Greece
fotiadis@uoi.gr

*Abstract*—**Deep learning has significantly progressed the field of medical image segmentation; yet, its susceptibility to adversarial attacks affects clinical and end-user confidence in the automated solutions derived from the AI systems. This study examines the adversarial robustness of five deep learning-based segmentation models and six loss functions against the Fast Gradient Sign Method attack across multiple attack strengths on a prostate MRI dataset. The experimental findings indicate that Recurrent U-Net had the highest adversarial robustness. Specifically, it surpassed all the other assessed models in two out of three evaluation metrics, achieving a mean of 0.52 Dice Coefficient and a 95th percentile Hausdorff Distance of 11.12 across the folds. Additionally, in the hold-out dataset, it attained a mean of 0.54 Dice Coefficient, a 95th percentile Hausdorff Distance of 9.45 and an Average Surface Distance of 0.93. Likewise, the loss functions derived from the Tversky index had the highest adversarial robustness. Precisely, Tversky loss surpassed all the other assessed loss functions in two out of three metrics across the folds, with a mean of 0.57 Dice Coefficient and 13.07 for the 95th percentile Hausdorff Distance and in all three evaluation metrics in the hold-out dataset, with a mean of 0.52 Dice Coefficient, a 95th percentile Hausdorff Distance of 10.05 and an Average Surface Distance of 0.74 when combined with the Binary Cross-Entropy loss function. From a clinical perspective, the findings of this work can guide the development of more adversarially resilient AI segmentation systems.**

*Index Terms*—**Adversarial robustness, Adversarial Attacks, Medical Image Segmentation, Prostate MRI, Deep Learning**

## I. INTRODUCTION

Prostate segmentation is a very important task in the diagnosis of prostate cancer in men. Deep learning-based segmentation models have significantly advanced this process by providing accurate delineation of the prostate gland and potential lesions from MRI, assisting doctors in their decision-making process. Despite their capabilities, their susceptibility to adversarial attacks raises significant concerns about the reliability of their functioning under this critical medical application. These attacks can modify the image such that the human eye cannot distinguish the differences between the original and the modified image [1]. Nevertheless, the inherent characteristics of AI models indicate that these minor modifications can affect this automated segmentation process.

The adversarial robustness of deep learning models is extensively studied, with various adversarial attacks having been proposed in the literature [2]. In the domain of medical imaging segmentation, there are several studies that examine the effects of the adversarial attacks on the performance of segmentation models [3], [4]. Nevertheless, these studies have been focused only on the adversarial robustness of the AI models themselves. Conversely, the role of the loss functions in adversarial robustness remains underexplored. The selected loss function is very crucial for an AI model's ability to learn to differentiate between the classes [5], but it can also negatively influence the adversarial robustness of the AI model itself.

Therefore, to our knowledge, this is the first work that evaluates the adversarial robustness of both the segmentation models and the loss functions, overall assessing the adversarial robustness of the main components within an AI segmentation system that may be utilized by the doctors for the automated segmentation solutions that they provide to their patients. The results demonstrate that segmentation models that incorporate recurrent convolutional layers into their original configuration and loss functions based on the Tversky index can significantly resist the adversarial noise.

The objective of this work was to conduct an adversarial robustness stress test on the state-of-the-art segmentation model architectures and loss functions that are widely used in the field of medical imaging segmentation, overall providing practical recommendations on selecting architectural components that are robust to the noise generated from the FGSM adversarial

attack, as it may enhance patient safety in scenarios where AI-generated segmentations are being employed. Moreover, the development of AI tools that exhibit reduced adversarial vulnerability can elevate doctors' confidence regarding the reliability of the automated segmentations produced by the AI systems that they utilize.

## II. MATERIALS & METHODS

### A. Imaging Dataset

The ProstateX [6] dataset was utilized for the experiments. This dataset comprises 3D T2-weighted magnetic resonance imaging (MRI) scans of the prostate from a total of 204 patients. It also contains binary segmentation masks of the whole gland, annotated by experienced radiologists. This dataset is part of the PICAI Challenge [7] and it is publicly available.

### B. Preprocessing steps

A series of preprocessing steps were performed to ensure uniformity in the shape and voxel intensity distribution of all MRIs. First, a bias correction filter was applied to address intensity variations, and the z-score normalization technique was employed on the images in order to standardize the voxel values. Furthermore, all the images and their corresponding masks were standardized to a fixed image shape of 256x256x1. A resampling operation for the voxel spacing was conducted by setting the spacing between neighboring voxels to 0.5x0.5x3.0, using linear interpolation. Finally, a Gaussian smoothing filter ($\sigma = 1.0$) was employed in order to suppress the high-frequency components of the images that arise due to hardware limitations or poor light conditions during the acquisition process of the MRIs.

### C. Deep Learning Models

In this work, we evaluated the adversarial robustness of five deep learning segmentation models. (a) The U-Net [8] model, which introduces a symmetrical U-shaped structure that consists of a downsampling path (encoder) and an upsampling path (decoder), where the feature maps produced by the encoder are concatenated with the corresponding feature maps from the decoder. (b) The Attention U-Net [9] which introduces attention gates (AGs) into the original U-Net decoder before the concatenation of the feature maps from the skip connections, assists the model in focusing on the most important regions of the input image. (c) The Recurrent U-Net [10], which replaces the standard convolutional layers with recurrent convolutional layers (RCLs), inputs the output of a layer at a given iteration back as an input to the same layer in the subsequent iteration. (d) The ResU-Net [11] introduces residual connections from the Residual Networks (ResNets) [12], in both the encoder and decoder parts of the U-Net model, enabling the training of deeper networks. (e) The U-Net combined with a Vision Transformer incorporates the ViT [13] in the deepest part (bottleneck) of the U-Net. The ViT tokenizes the input image into a sequence of patches, where multiple self-attention and feedforward neural network layers are then used to capture their global information.

### D. Loss Functions

The models presented in Section II-C are trained using the following loss functions, both individually and in weighted combinations: (a) The Binary Cross-Entropy (BCE) loss is a widely used loss function specifically for binary classification tasks. It measures the difference between the probability of the predicted pixel/voxel belonging to the target and the actual ground truth class. (b) The Dice loss [14] is based on the Dice Coefficient (DC), which measures the overlap between the predicted and ground truth segmentation masks. (c) The Tversky loss [15] is a generalised version of the Dice loss, which controls the trade-off between false positives and false negatives. In addition, we also took the weighted combinations of the previous loss functions. (d) The Binary Cross Entropy-Dice loss (BCE-Dice), which is a loss function that is the weighted sum of the BCE and the Dice loss [16]. (e) The Binary Cross Entropy-Tversky (BCE-Tversky) loss function, which combines the BCE and Tversky loss functions as a weighted sum [16]. (f) The Dice-Tversky loss function, which occurs by taking the weighted sum of the Dice and Tversky loss functions [16].

### E. Adversarial Attack Method

The adversarial robustness assessment of the segmentation models and loss functions was performed by employing the Fast Gradient Sign Method (FGSM) attack [1], which is a computationally efficient one-step adversarial attack. FGSM is a white-box attack, meaning that the attacker has complete knowledge of the target model, including its architecture, parameters, and gradients. It operates by calculating the gradient of the model's loss function with respect to the input and then modifies the input in the direction of the sign of this gradient, overall maximizing the model's likelihood of misclassification.

The adversarial image $\mathbf{x_{adv}}$ is generated from the original input $\mathbf{x}$ using the following equation:

$$\mathbf{x_{adv}} = \mathbf{x} + \epsilon \cdot \text{sign}(\nabla_{\mathbf{x}}\mathcal{L}(\boldsymbol{\theta}, \mathbf{x}, y)) \quad (1)$$

where $\mathbf{x}$ is the original input image, $y$ is the ground truth label corresponding to $\mathbf{x}$, $\mathcal{L}(\theta, \mathbf{x}, y)$ is the loss function used to train the model with parameters $\boldsymbol{\theta}$, $\nabla_{\mathbf{x}}\mathcal{L}(\boldsymbol{\theta}, \mathbf{x}, y)$ is the gradient of the loss function with respect to the input image $\mathbf{x}$, $\text{sign}(\cdot)$ is the sign function, and $\epsilon$ controls the attack strength.

In our experiments, the FGSM attack was applied to each image in both the cross-validation phase and the hold-out dataset (Section II-G), using a range of attack strengths. Specifically $\epsilon \in \{0.1, 0.2, \ldots, 1.0\}$. While the goal of the adversarial attacks is to generate adversarial examples imperceptible to the human eye, the selection of the values for the adversarial strength ($\epsilon$) was perfomed with the intention to produce not only weak but also strong enough adversarial examples, to test the limits of the presented model architectures and loss functions under a potential adversarial attack.

### F. Evaluation Metrics

Three evaluation metrics were utilized in the present work. (a) The Dice Coefficient [17], which measures the similarity

between two given samples (in this case the predicted and ground truth segmentation masks). It quantifies how much the two samples overlap. (b) The 95th percentile Hausdorff Distance (HD95), which is a distance metric that is an extension of the original Hausdorff Distance (HD) [17] that measures the maximum spatial distance between the predicted and ground truth segmentation boundaries, with the difference being that the HD95 excludes the largest 5% of the distances, making it more robust to outliers. (c) The Average Surface Distance (ASD) [17] which is also a distance metric that calculates the average symmetric distance between the predicted and ground truth segmentation boundaries.

### G. Experimental Design

The ProstateX dataset, which consists of 204 patients, was initially split into a training set (70%, 143 patients), a validation set (15%, 30 patients), and a hold-out test set (15%, 31 patients). Then the following two evaluation phases were conducted:

*Cross-Validation*: A 3-fold cross-validation was conducted within the 70% training set, where the FGSM attack was applied to each validation fold, and the results that are presented for the models in Table I and for the loss functions in Table III of Section III represent the average results across the folds and adversarial attack strengths ($\epsilon$).

*Hold-Out Dataset*: After cross-validation, the models were retrained with the loss functions from Section II-D on the entire 70% training dataset. The final robustness evaluation was performed by applying the FGSM attack to the independent 15% hold-out dataset. The results that are presented for the models in Table II and for the loss functions in Table IV of Section III represent the average results across all the adversarial attack strengths ($\epsilon$) values which are presented in Section II-E.

The two described evaluation phases have been conducted in order to ensure that the experimental results regarding the adversarial robustness are consistent and representative for all the model architectures and loss functions. It should be mentioned that based on the methodological description, a standard arithmetic mean of the results was used. Specifically, all the adversarial strength values ($\epsilon$) for the FGSM attack were treated with equal importance for the final calculation of the results for the models and loss functions for both the cross-validation strategy and the hold-out dataset.

### H. Training and architecture parameters

All segmentation models were trained using the TensorFlow 2 framework on an NVIDIA A40 GPU for up to 150 epochs, using a batch size of 4. The Adam optimizer [18] was selected, with an initial learning rate (LR) of $1 \times 10^{-4}$. Furthermore, a reduction scheduling of the initial LR was applied if the validation loss was not improving for 2 consecutive epochs. This reduction was continued until a minimum LR value of $1 \times 10^{-6}$ was reached. In addition, to prevent unnecessary overfitting to the training data, an early stopping was applied. To ensure a fair comparison on the unique components of

each model (e.g., attention gates, recurrent convolution layers, residual connections, etc.) we chose fixed encoder-decoder depth, number of filters per level and activation functions. The fixed selection of all the architectural parameters for the models, guarantees the non existence of bias against a specific model architecture.

### III. RESULTS

The segmentation models and loss functions were assessed based on the evaluation metrics presented in Section II-F. The results, which are presented in this section represent the average performance results across the 3-fold cross-validation that is performed on the training dataset and in the hold-out dataset, as discussed in Section II-G.

### A. Comparison of the different model architectures

Table I summarizes the models' average performance results for the cross-validation strategy after the FGSM attack was performed in each validation fold. Based on the results, Recurrent U-Net achieved the highest mean DC and lowest mean HD95, suggesting adversarial robustness due to the recurrent convolutional layers. On the other hand, while Attention U-Net achieved a competitive mean ASD compared to Recurrent U-Net, its higher standard deviation suggests Recurrent U-Net was more consistent in this metric compared to Attention U-Net. Conversely, based on the evaluation metrics, the ResU-Net achieved the least adversarial robustness performance compared to the rest of models.

TABLE I
MEAN RESULTS AND STANDARD DEVIATIONS ACROSS FOLDS FOR EACH MODEL AFTER THE FGSM ATTACK IS APPLIED IN EACH VALIDATION FOLD

| Model | DC | HD95 (mm) | ASD (mm) |
|---|---|---|---|
| U-Net | $0.49 \pm 0.17$ | $13.33 \pm 7.79$ | $1.72 \pm 2.84$ |
| Attention U-Net | $0.48 \pm 0.21$ | $12.60 \pm 7.58$ | $\mathbf{1.43 \pm 3.23}$ |
| Recurrent U-Net | $\mathbf{0.52 \pm 0.11}$ | $\mathbf{11.12 \pm 5.53}$ | $1.45 \pm 2.03$ |
| ResU-Net | $0.35 \pm 0.32$ | $36.17 \pm 39.99$ | $33.03 \pm 42.73$ |
| UNet-ViT | $0.5 \pm 0.14$ | $13.59 \pm 6.57$ | $2.28 \pm 2.76$ |

Similarly, Table II shows the average results for the hold-out dataset for all the models. It is evident that Recurrent U-Net again attained the best mean results across all evaluation metrics, indicating the advantages of introducing recurrent convolutional layers into the original U-Net architecture, which shows adversarial robustness against the FGSM attack.

TABLE II
MEAN RESULTS AND STANDARD DEVIATION ACROSS THE HOLD-OUT DATASET FOR EACH SEGMENTATION MODEL AFTER THE FGSM ATTACK IS APPLIED IN THE HOLD-OUT DATASET

| Model | DC | HD95 (mm) | ASD (mm) |
|---|---|---|---|
| U-Net | $0.52 \pm 0.22$ | $10.43 \pm 3.77$ | $1.2 \pm 1.31$ |
| Attention U-Net | $0.48 \pm 0.17$ | $16.17 \pm 16.81$ | $4.19 \pm 9.23$ |
| Recurrent U-Net | $\mathbf{0.54 \pm 0.1}$ | $\mathbf{9.45 \pm 2.93}$ | $\mathbf{0.93 \pm 0.42}$ |
| ResU-Net | $0.37 \pm 0.28$ | $25.9 \pm 33.57$ | $15.1 \pm 27.45$ |
| UNet-ViT | $0.51 \pm 0.21$ | $14.61 \pm 11.06$ | $3.61 \pm 7.52$ |

Figure 1 presents the segmentation masks derived from the ResU-Net under the FGSM attack, when it is trained with the most robust loss function from Section III-B. The difference heatmaps visualize the pixel values distribution of the added

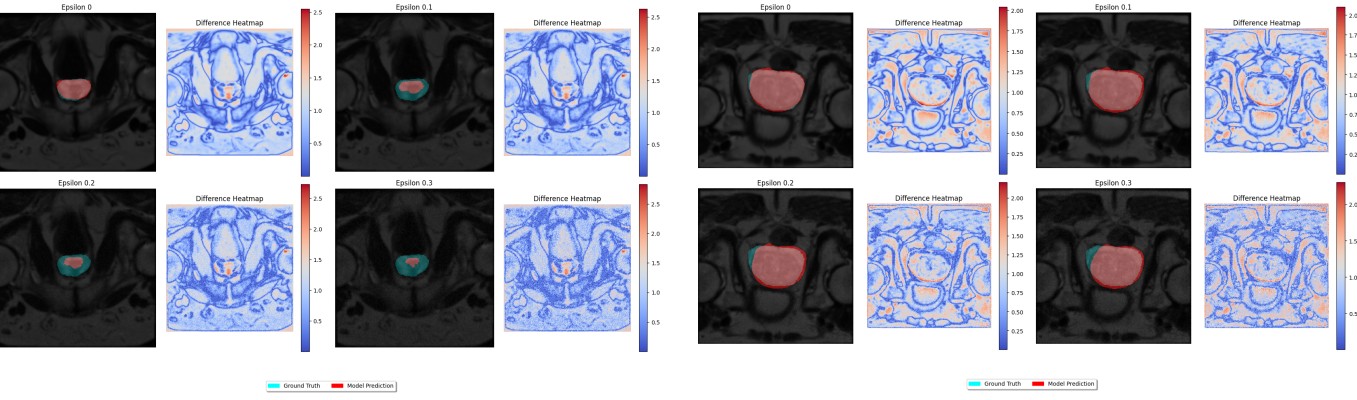

Fig. 1. Segmentation performance of the ResUNet on a random MRI slice under the FGSM attack. As the adversarial strength increase, it loses its robustness, overall producing low quality segmentation masks.

Fig. 2. Segmentation performance of the Recurrent U-Net under the FGSM attack on a random MRI slice. Compared to ResU-Net, as the adversarial strength increases, RecurrentU-Net is more robust overall maintaining its ability to generate accurate prostate segmentation masks.

adversarial noise to the clean MRI slice. It is evident that the quality of the predicted segmentation masks degrades as the adversarial strength increases, where $\epsilon = 0$ represents the clean MRI slice, when no adversarial noise is added. The visual representations are supported by the experimental findings from Table I and Table II, where the ResU-Net consistently achieves the lowest mean DC and the highest mean values for the distance metrics among the rest of the models across the folds and the hold-out dataset, indicating that the specific architecture is more vulnerable to the FGSM attack, even when it is trained with the most robust loss function.

Conversely, Figure 2 illustrates the generated segmentation masks from the Recurrent-UNet under the FGSM attack on a random MRI slice when it is trained with the most robust loss function from Section III-B. As illustrated in Figure 2, the RecurrentU-Net is more capable of maintaining its adversarial robustness as the adversarial attack strength ($\epsilon$) increases, compared to the ResU-Net. The superior performance of the Recurrent U-Net can be further supported by the experimental results from Table I and Table II, where it achieves the highest mean DC and the lowest mean value for the 95HD among the rest of the model architectures across the folds and the hold-out dataset.

### B. Comparison of the different loss functions

Table III presents the average cross-validation results for the loss functions. Tversky loss attained the highest mean DC and the lowest mean HD95, indicating its effectiveness in controlling the trade-off between false positives and negatives. While Dice loss showed the lowest mean ASD, its higher standard deviation compared to Tversky suggests Tversky provided a more consistent average symmetric distance of the surfaces compared to the Dice loss.

Table IV shows the average results of the loss functions on the hold-out dataset. The combination of the BCE and Tversky demonstrated the highest levels of adversarial robustness. It is evident that when the models are trained with the Tversky loss, either standalone (Table III) or combined (Table IV), they provide enhanced adversarial robustness levels, which

TABLE III
MEAN RESULTS AND STANDARD DEVIATIONS ACROSS THE FOLDS FOR EACH LOSS FUNCTION AFTER THE FGSM ATTACK IS APPLIED IN EACH VALIDATION FOLD

| Loss Function | DC | HD95 (mm) | ASD (mm) |
|---|---|---|---|
| BCE | 0.37 ± 0.21 | 17.21 ± 11.96 | 12.09 ± 29.58 |
| Dice | 0.49 ± 0.21 | 14.95 ± 15.59 | **3.37 ± 11.03** |
| Tversky | **0.57 ± 0.16** | **13.07 ± 13.06** | 3.57 ± 7.39 |
| BCE-Dice | 0.5 ± 0.26 | 14.04 ± 16.4 | 6.38 ± 18.64 |
| BCE-Tversky | 0.46 ± 0.23 | 23.41 ± 34.87 | 15.07 ± 36.68 |
| Dice-Tversky | 0.44 ± 0.22 | 21.47 ± 23.6 | 7.37 ± 17.74 |

underscores the potential benefit of controlling the trade-off between false positives and false negatives.

TABLE IV
MEAN RESULTS AND STANDARD DEVIATIONS FOR THE HOLD-OUT DATASET FOR EACH LOSS FUNCTION AFTER THE FGSM ATTACK IS APPLIED IN THE HOLD-OUT DATASET

| Loss Function | DC | HD95 (mm) | ASD (mm) |
|---|---|---|---|
| BCE | 0.43 ± 0.21 | 12.24 ± 5.15 | 2.3 ± 4.96 |
| Dice | 0.47 ± 0.18 | 16.76 ± 18.36 | 5.07 ± 11.51 |
| Tversky | 0.5 ± 0.15 | 15.43 ± 13 | 4.33 ± 7.6 |
| BCE-Dice | 0.49 ± 0.19 | 14.63 ± 19.22 | 6.97 ± 19.44 |
| BCE-Tversky | **0.52 ± 0.1** | **10.05 ± 5.88** | **0.74 ± 5.8** |
| Dice-Tversky | 0.49 ± 0.17 | 23.49 ± 31.96 | 10.85 ± 24.12 |

Figure 3 illustrates the poor segmentation performance of the Recurrent U-Net when the BCE loss function is used for its training. It is evident that as the adversarial strength increases, the quality of the model's prediction degrades. Specifically, the overlap between the predicted and the ground truth prostate segmentation masks is minimized. The degradation performance of the BCE is also highlighted from the experimental results presented in Table III and Table IV, where it achieves the lowest mean DC among the rest of the evaluated loss functions.

### IV. DISCUSSION

The experimental results demonstrated the enhanced performance of the Recurrent U-Net model architecture and loss functions that are based on the Tversky index against the FGSM attack. Specifically, the Recurrent U-Net achieved in

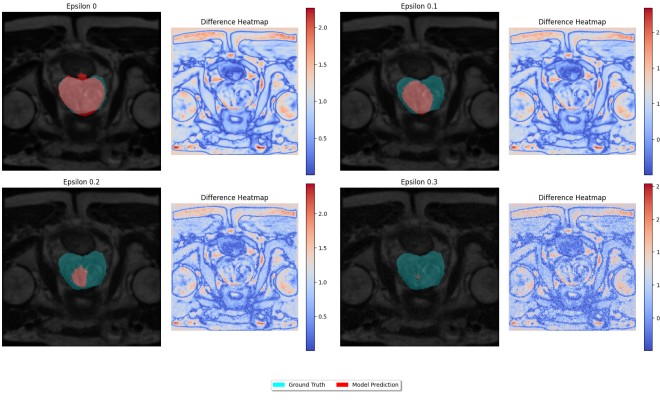

Fig. 3. Segmentation performance of the Recurrent U-Net when the BCE loss function is used for its training. As the adversarial strength ($\epsilon$) increases, the segmentation performance of the model degrades.

the cross-validation phase (Table I) mean 52% Dice Coefficient (DC) and 54% in the hold-out dataset (Table II). In contrast, ResU-Net showed poor levels of adversarial robustness, achieving mean 35% DC in the cross-validation phase (Table I) and 37% in the hold-out dataset (Table II), suggesting that despite residual connections being an efficient technique for training deeper networks, they do not automatically guarantee adversarial robustness against the FGSM attack. The performance gap between the Recurrent U-Net and Res U-Net exists due to their fundamental differences in their architectural components. Recurrent U-Net introduces recurrent connections, while Res U-Net introduces residual connections. In particular, recurrent connections can process the feature maps multiple times, allowing the model to filter the noise that is presented in the image. Conversely, residual connections introduce a shortcut connection that adds the input of a block into its output [12]. Although this mechanism has been proven to be very effective in training deeper networks, because they can efficiently mitigate the problem of vanishing/exploding gradients [19], it can unintentionally make the network more vulnerable to the adversarial attacks, as the residual connections can propagate the adversarial noise more effectively throughout the entire network.

The Tversky loss function has shown, both independently in the cross-validation phase (Table III), with a mean 57% DC and in conjunction with BCE in the hold-out dataset (Table IV), with mean 52% DC, the importance of controlling the trade-off between false positives and false negatives. Conversely, the standard BCE showed the lowest levels of adversarial robustness, only achieving mean 37% DC during the cross-validation phase (Table III) and mean 43% DC in the hold-out dataset, indicating higher vulnerability to the FGSM attack in this task. As confirmed by the experimental findings (Section III-B), the significant difference in the adversarial robustness levels between the models trained with Tversky and BCE loss functions is a result of their fundamental difference in their pixel processing operations. Specifically, the Tversky is a region-based loss and was developed with the goal to address the problem of high class imbalance that occurs in

the field of medical imaging segmentation, by controlling the trade-off between false positives and false negatives [15]. Conversely, BCE operates on individual pixels. As a result, it treats all pixels with equal significance, which can cause the model to put significance into the majority class (background), leading to poor detection of the small regions of the image (foreground).

Our work aligns with existing research that explores the vulnerabilities of deep learning models in medical imaging segmentation [3], [4]. Previous studies have focused mainly on either the model architectures [3], [4] or the loss functions [5], [14], [16] without quantifying their adversarial robustness. However, a novel aspect of this work is that it evaluates the adversarial robustness of both model architectures and loss functions, illustrating that, in addition to the model architecture, the careful consideration of a robust loss function is essential for developing an adversarially resilient AI segmentation system.

Accurate prostate segmentation from medical images is essential for the diagnosis of prostate cancer in men. It is essential that segmentations produced by AI systems are accurate and robust to potentially minor modifications in MRIs in order for the doctors to be confident that they deliver the best treatment to their patients.

A limitation of this work is its focus on a single one-step, white-box adversarial attack and a single dataset. Consequently, the observed robustness levels may differ when models and loss functions are subjected to iterative attacks or when applied to datasets with different imaging modalities, such as images from X-rays, CT scan or ultrasound. Nevertheless, these findings underscore the impact of selecting model architectures and loss functions that are robust to the FGSM attack.

## V. Conclusion

The purpose of this paper is to assess the adversarial robustness of five segmentation models and six loss functions widely used in medical imaging segmentation against the FGSM attack on a prostate MRI dataset. The results demonstrate that the recurrent convolutional layers and loss functions that incorporate the Tversky index enhance the adversarial robustness. Future work will aim to investigate the presented configurations against multi-step adversarial attacks and validate the generalizability of the results across additional prostate MRI datasets and in datasets with different medical imaging modalities, such as images from X-ray, CT scan and ultrasound.

### Acknowledgment

This work is funded by the European Union's Horizon FAITH project (Fostering Artificial Intelligence Trust for Humans towards the optimization of trustworthiness through large-scale pilots in critical domains), Grant Agreement No. 101135932. It reflects only the authors' view and the European Commission is not responsible for any use that may be made of the information it contains.

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
