# OpenReview forum: "Adversarial Robustness Evaluation of Deep Learning Segmentation Models and Loss Functions in Prostate MRI"
_IEEE.org/EMBS/BHI/2025/Conference — BHI 2025_

### Official Review · Reviewer_Ut5e · 2025-07-15
**Good evaluation**

**Confidence:** 2
**Clarity Of Writing:** good
**Clinical Significance:** good
**Methodological Novelty:** fair
**Overall Rating:** 5

**Experiments And Results:**

fair

**Questions For The Authors:**

1.The choice of attack strength range (ε ∈ {0.1, 0.2, ..., 1.0}) appears arbitrary without justification.
2. Can you provide analysis of perceptual quality and whether such perturbations could realistically occur in clinical settings?

**Strengths:**

The simultaneous evaluation of both model architectures and loss functions for adversarial robustness is genuinely novel and provides comprehensive insights into system-level robustness factors. Moreover, the study includes multiple model architectures, loss functions, and a proper evaluation methodology with both cross-validation and hold-out testing across multiple attack strengths.

**Summary Of The Paper:**

This study evaluates the adversarial robustness of five deep learning segmentation models (U-Net, Attention U-Net, Recurrent U-Net, ResU-Net, and UNet-ViT) and six loss functions (BCE, Dice, Tversky, and their combinations) against Fast Gradient Sign Method (FGSM) attacks on prostate MRI segmentation.

**Weaknesses:**

1. The evaluation is restricted to a single white-box attack (FGSM), which is computationally efficient but relatively simple.

---

### Official Review · Reviewer_CLM7 · 2025-07-15
**A Comparative Analysis of Segmentation Models and Loss Functions in Medical Imaging Against FGSM Attacks**

**Confidence:** 4
**Clarity Of Writing:** great
**Clinical Significance:** good
**Methodological Novelty:** good
**Overall Rating:** 7

**Experiments And Results:**

excellent

**Questions For The Authors:**

Does your findings can be confirmed with another dataset?

**Strengths:**

A clear experiment to assess the adversarial robustness of five segmentation models and six loss functions widely used in medical imaging segmentation against the FGSM attack on a prostate MRI dataset makes it repeatable by others.

**Summary Of The Paper:**

The purpose of this paper is to assess the adversarial robustness of five segmentation models and six loss functions widely used in medical imaging segmentation against the FGSM attack on a prostate MRI dataset. The authors' experiment highlights that recurrent convolutional layers and loss functions incorporating the Tversky index enhance adversarial robustness.

**Weaknesses:**

This paper does not introduce a new segmentation method that limits its innovation. However, a clear experiment design can inform others on the robustness of segmentation methods and loss functions against the FGSM attack.

---

### Official Review · Reviewer_DPh1 · 2025-07-17
**49**

**Confidence:** 3
**Clarity Of Writing:** good
**Clinical Significance:** good
**Methodological Novelty:** fair
**Overall Rating:** 4

**Experiments And Results:**

good

**Questions For The Authors:**

Please refer to the weaknesses listed above for details.

**Strengths:**

1, Evaluates both model architecture and loss function, providing a comprehensive view of robustness factors.\
2, Experiments are conducted on a well-known, publicly available dataset (ProstateX), enhancing reproducibility.\
3, Includes both cross-validation and hold-out test set results, reinforcing the credibility of the findings.

**Summary Of The Paper:**

This paper evaluates the adversarial robustness of five deep learning segmentation models and six loss functions under FGSM attacks in prostate MRI segmentation.

**Weaknesses:**

1, Only one dataset is used (ProstateX), with no validation on other anatomical sites or imaging modalities.\
2, The visual analysis is lacking—no qualitative segmentation maps or visual examples of adversarial perturbations are shown.\
3, Model implementation details (e.g., optimizer parameters, training epochs, learning rate) are missing or insufficiently described.\

---

### Official Review · Reviewer_vpUB · 2025-07-17
**This study investigates the adversarial robustness of five deep learning segmentation models and six loss functions against the Fast Gradient Sign Method (FGSM) attack on a prostate MRI dataset. The key findings indicate that Recurrent U-Net demonstrated the highest adversarial robustness among the models, and Tversky-based loss functions (Tversky and BCE-Tversky) showed superior robustness among the loss functions.**

**Confidence:** 5
**Clarity Of Writing:** great
**Clinical Significance:** great
**Methodological Novelty:** great
**Overall Rating:** 7

**Experiments And Results:**

good

**Questions For The Authors:**

- The paper states that ϵ values ranging from 0.1 to 1.0 were used for attack strengths, and the final metrics were averaged across these values. To account for potential differences in impact and importance across ϵ values, was a weighted average used, where each output corresponding to an ϵ value was assigned a weight based on the corresponding ϵ relative importance?
- Table I: "... DEVIATIONS ACROSS FOLDS FOR EACH ...": "... DEVIATIONS ACROSS FOLDS and attack strengths (\epsilon) FOR EACH ..." (is the averaging across folds and attack strengths?)

**Strengths:**

- The adversarial robustness assessment was performed on both segmentation models and loss functions.
- Clinical relevance
- The usage of a diverse set of commonly used deep learning segmentation models and a variety of loss functions

**Summary Of The Paper:**

The authors utilized the ProstateX dataset, comprising 3D T2-weighted MRI scans and expert-annotated segmentation masks, five deep learning models (U-Net, Attention U-Net, Recurrent U-Net, ResU-Net, and U-Net combined with a Vision Transformer (UNet-ViT)), and six loss functions (Binary Cross-Entropy (BCE), Dice loss, Tversky loss, and their weighted combinations (BCE-Dice, BCE-Tversky, Dice-Tversky)) to assess the adversarial robustness using the Fast Gradient Sign Method (FGSM) attack with varying strengths (ϵ∈{0.1,0.2,...,1.0}) using the evaluation metrics of Dice Coefficient (DC), 95th percentile Hausdorff Distance (HD95), and Average Surface Distance (ASD). The results demonstrated that Recurrent U-Net consistently outperformed other models.

**Weaknesses:**

- The study uses a single dataset (ProstateX) and attack mechanism (FGSM), which limits the generalizability of the findings.

---

### Official Review · Reviewer_aMWq · 2025-07-21
**The paper provides a review of a range of deep learning networks and loss functions for prostate MRI data.**

**Confidence:** 4
**Clarity Of Writing:** good
**Clinical Significance:** good
**Methodological Novelty:** fair
**Overall Rating:** 5

**Experiments And Results:**

fair

**Questions For The Authors:**

See above.

**Strengths:**

Evaluation is good, but could include an estimation of the statistical significance of the differences.

**Summary Of The Paper:**

The paper provides a review of a range of deep learning networks and loss functions for prostate MRI data.

**Weaknesses:**

Further discussion of the results would be useful and could provide additional insight.